



**Identification of new microbial functional standards for soil quality assessment**
**Sören Thiele-Bruhn [1], Michael Schloter [2], Berndt-Michael Wilke [3], Lee A. Beaudette [4], Fabrice Martin-**
**Laurent [5], Nathalie Cheviron [6], Christian Mougin [6], Jörg Römbke [7]**
[1] Universität Trier, Bodenkunde, Behringstr. 21, 54286 Trier, Germany
[2] Helmholtz Zentrum München, Deutsches Forschungszentrum für Gesundheit und Umwelt, Abteilung für
vergleichende Mikrobiomanalysen, Ingolstädter Landstr. 1, 85764 Neuherberg, Germany
[3] TU Berlin, FG Bodenkunde, Ernst-Reuter-Platz 1, 10587 Berlin, Germany
[4] Environment and Climate Change Canada, 335 River Road, Ottawa, Ontario, K1A 0H3, Canada
[5]AgroSup Dijon, INRA, Université Bourgogne, Université Bourgogne Franche-Comté, Agroécologie, 17 rue
Sully, 21065 Dijon Cédex, France
[6] UMR ECOSYS, Platform Biochem-Env, INRA, AgroParisTech, Université Paris-Saclay, 78026, Versailles,
France
[7] ECT Oekotoxikologie GmbH, Böttgerstr. 2-14, 65439 Flörsheim, Germany
**Correspondence:** Sören Thiele-Bruhn (thiele@uni-trier.de)



**Abstract.** The activity of microorganisms in soil is important for a robust functioning soil and related
ecosystem service. Hence, there is a necessity to identify the indigenous soil microbial community for its
functional properties using soil microbiological methods in order to determine the natural properties,
functioning and operating range of soil microbial communities, and to assess ecotoxicological effects due
to anthropogenic activities. Numerous microbiological methods currently exist in the literature and new,
more advanced methods continue to be developed; however, only a limited number of the methods are
standardized. Consequently, there is a need to identify the most promising non-standardized methods for
assessing soil quality and develop these into standards. In alignment with the "Ecosystem Service
Approach", new methods should focus on soil microbial function, including nutrient cycling, pest control
and plant growth promotion, carbon cycling and sequestration, greenhouse gas emission, and soil
structure. The few existing, function-related standard methods available focus on the estimation of
microbial biomass, basal respiration, enzyme activities related to nutrient cycling, and organic chemical
biodegradation. This paper sets out to summarize and expand on recent discussions within the
International Organization for Standardization (ISO), Soil Quality - Biological Characterization sub-
committee (ISO TC 190/SC 4) where a need was identified to develop scientifically sound methods which
would best fulfil the practical needs of future users for assessing soil quality. Of particular note was the
current evolution of molecular methods in microbial ecology that uses quantitative real time PCR (qPCR)
to produce a large number of new endpoints and is more sensitive as compared to 'classical' methods.
Quantitative PCR assesses the activity of microbial genes that code for enzymes that catalyse major
transformation steps in nitrogen and phosphorus cycling, greenhouse gas emissions, chemical
transformations including pesticide degradation, and plant growth promotion pathways. In the
assessment of soil quality methods, it was found that fungal methods were significantly underrepresented.
As such, techniques to analyse fungal enzyme activities are proposed. Additionally, methods for the
determination of microbial growth rates and efficiencies, including the use of glomalin as a biochemical



marker for soil aggregation, are discussed. Furthermore, field methods indicative of carbon turnover,
including the litter bag test and a modification to the tea bag test, are presented. As a final note, it is
suggested that endpoints should represent a potential function of soil microorganisms rather than actual
activity levels, as the latter can largely be dependent on short-term variable soil properties such as
pedoclimatic conditions, nutrient availability, and anthropogenic soil cultivation activities.



## 1 Introduction

Soils are one of the world's hotspots for biodiversity (Parker, 2010). Biota – both micro- and macro-organisms - in soil form strong networks and complex food webs, which determines the efficacy of the soil ecosystem functions (e.g. nutrient cycling, C storage and turnover, water retention, and modulation of soil structure) (Creamer et al., 2016). These functions support a range of ecosystem services that are indispensable for soil use in agri-, horti- or silviculture (Nannipieri et al., 2017). At the same time, soil biota are strongly impacted by various anthropogenic activities including ongoing global and climate change, pollution, and degradation and destruction of the terrestrial environment (Gomiero, 2016; Montgomery, 2008; Wagg et al., 2014). Consequently, investigations of the soil biome structure and function became an emerging topic in soil and environmental sciences (Griffiths and Philippot, 2013). As such, the number of publications on soil ecology and ecosystem functioning has increased significantly over the past few decades and has resulted in the development of new methods (e.g. Guillaume et al., 2016; Tian et al., 2018). In comparison, the ecotoxicological assessment of human impacts (e.g. chemical pollution and mechanical compaction) using single species tests, which are well-established methods, has remained constant (Brookes, 1995; Joergensen and Emmerling, 2006).

Characterizing the natural state of a soil's biome is quite a challenging task. In addition to its huge structural and functional diversity, the soil biome is influenced by strong temporal dynamics including seasonal weather conditions and the enormous spatial heterogeneity which ranges from field scale to microscale (Kuffner et al., 2012; Regan et al., 2014; Suriyavirun et al., 2019). All of these intrinsic properties hinder the interpretation of data obtained from the analysis of soil biomes and the measurement of their functional traits.

Despite the fundamental methodological advances over the past years, which allow for an in-depth analysis of microbiomes and, to some extent, other soil-living organisms (e.g. Joergensen and Emmerling, 2006; Paul, 2015; Yates et al., 2016), only a limited number of soil biological methods have





been standardized (for details see section 3). As a result, large deviations are observed between non-
standardized method protocols (e.g. Strickland and Rousk, 2010). Therefore, comparability between
datasets generated by different laboratories using different methods or modified protocols of the same
method are problematic. Consequently, the development of quality indices and threshold values,
respectively, for assessing soil quality is nearly impossible (Bastida et al., 2008). Presumably, this is why
the number of meta-analyses in soil biology remains small.

Given that there is a lack of harmonization between existing methods and, at the same time, a

proliferation of new methods, there is a need to identify the most promising methods described in the
literature that can be standardized to produce reliable indicators for soil quality (e.g. Philippot et al., 2012).
At the Annual International Organization for Standardization (ISO) meeting of TC 190 (Soil Quality) in
Fukuoka, Japan in October 2013, a decision was made to compile a list of available methods and to identify
those that would be suitable for assessing soil quality. Additionally, during a subsequent meeting of ISO
TC 190/SC 4/WG 4 (Microbiological Methods) held in Paris, France in March 2014, further discussions
focused the criteria for suitable methods to be comprised of microbial functional indicators. In this paper,
we summarize the major outcomes of the discussions which took place over the past several years within
ISO TC 190/SC 4. Therefore, besides collating a list of criteria for the selection of test methods for the
future analysis of microbial functions in soil, the aim of this paper is to present our opinion, as members
of the ISO TC 190 committee, to initiate further discussion on possible methods that should be
standardized for future soil quality assessments.

**2 Criteria for the selection of methods**
Several papers addressing the task to identify suitable methods to be used as biotic indicators (usually
including faunal indicators) were published in the last few years, mainly in the context of EU research
projects (e.g. Bispo et al., 2009; Faber et al., 2013; Ritz et al., 2009; Römbke et al., 2010). Here, we propose





to base the selection of soil quality methods more on the "Ecosystem Service Approach" (MEA 2005) which
is increasingly recognized by both environmental scientists and regulatory agencies (Breure et al., 2012;
Galic et al., 2012) and that soils have been raised to the rank of a natural resource to be protected. As a
consequence, and in addition to method development and application (including the assessment of
biodiversity as a prerequisite for soil function), the focus of future activities should be the determination
of soil microbial function as recommended endpoints (Kvas et al., 2017; Nienstedt et al., 2012; van der
Putten et al., 2010; TEEB, 2010). Consequently, we propose to assess both existing and new methods for
the selection of microbial functional tests that support various soil ecosystem services. This structures our
approach and simplifies the identification of ecologically relevant methods, as well as, presumably
increasing their acceptance by users, including the regulatory and stakeholder community. The following
soil **Functions** and ecosystem services have been defined and are proposed to be used as a starting point
for the development of future methods (MEA, 2005; Ockleford et al., 2017):
(1) Biodiversity, genetic resources, cultural services;
(2) Food web support;
(3) Biodegradation of pollutants;
(4) Nutrient cycling (for example N and P);
(5) Pest control and plant growth promotion;
(6) Carbon cycling and sequestration;
(7) Greenhouse gas emissions; and
(8) Soil structure affecting soil water, gas balance and filtration function.
A second major criterion for selecting methods for standardization is its usability. The method should be
applicable in regulations (e.g. European and National agencies registering chemicals or products) and for
the evaluation of soil ecology and functioning as a fundamental aspects of soil quality (e.g. by stakeholders
and researchers). Moreover, the routine use of methods to inform farmers and site owners on soil quality



as continuous assessments of their land and land-use practises, could be an additional condition that
would require the choice of easy-to-use methods or possibly encourage the simplification of existing
methods. Frequently used methods generate more data, which in turn is of high importance for the
validation of threshold values.

To assess possible methods, a list of criteria was used based on the 'logical sieve' approach (Ritz

et al., 2009). The list of criteria for the identification of functional indicators and associated methodologies
(Table 1) was an outcome of the EU FP7 EcoFINDERS project (Faber et al., 2013). The criteria were compiled
after sending a questionnaire to 25 partner institutions primarily working in the field of environmental
science; mainly representing academia but also regulators and subcontracting laboratories. These criteria
are applicable for different kinds of indicators and methods, including those addressing the functions of
soil microbial communities. In the following sections, we assume that existing ISO standardized methods
already fulfil these criteria. Additionally, the most appropriate new methods, including those proposed in
this article, need to be evaluated using the same criteria required for the standardization of ISO methods.
Therefore, the aim of this whole process is to identify methods which are scientifically sound and that best
fulfil the practical needs of future users.

**3 Existing and new methods**
Current methods that have already been implemented as ISO standards are found in Table 2, whereas
methods that might be considered for future standardization are in Table 3. The compilation in Table 2
comprises methods to quantify microbial biomass (e.g. through fumigation extraction of microbial biomass
carbon (MBC) and DNA) (**Function 6**) as well as for (further) analysis of structural microbial diversity (e.g.
determination of microbial fingerprints by phospholipid fatty acids (PLFA) analysis) (**Function 1**).
Additionally, microbial biomass, measured as respiratory activity, has been included in Table 2, although
not directly linked to one of the ecosystem services, as it provides important information on the activity



of the complete microbiome (i.e. microflora and microfauna). Soil basal respiration normalized to MBC
(ISO 14240-1 and ISO 14240-2, 1997, Table 2) yields the metabolic quotient $qCO_2$, which is a sensitive
indicator for microbial carbon use efficiency (Anderson and Domsch, 1993). Interestingly, its use as an
endpoint to assess anthropogenic and natural impacts on the soil microbiome has been controversially
discussed in literature (Wardle and Ghani, 1995).
The biodiversity function (**Function 1**) addresses parameters related to the structural diversity of
the soil microbiome. Here, respective ISO guidelines analysing PLFA, phospholipid ether lipids (PLEL)
(ISO/TS 29843-1, 2010; ISO/TS 29843-2, 2011) and DNA (ISO 11063, 2012; ISO 17601, 2016), have already
been well implemented into guidelines (Table 2). Although microbial diversity, per se, is not strongly
correlated with a particular functional capacity, it is clear that the loss of diversity can have an impact on
microbial function (Thiele-Bruhn et al., 2012); at least for relatively specific functions performed by narrow
microbial guilds or taxa. This applies even more, when certain taxa are closely linked to very specific
functions including nitrifiers, methanogens, arbuscular- and ecto-mycorrhizal fungi, and biocontrol
microorganisms like *Trichoderma* (e.g. Hartmann et al., 2009; Hayat et al., 2010; Lugtenberg and Kamilova,
2009; Peng et al., 2008; Singh et al., 2007; Xia et al., 2011). Therefore, the interpretation of the outcomes
from microbial community-based testing tends to be straightforward and closely linked to **Function 4** and
**Function 5**.
Food web support (**Function 2**) of higher trophic levels no doubt starts from soil microorganisms
and propagates through the trophic levels (e.g. earthworms) that are consumed by birds and mammals
(Haynes, 2014; Scheu et al., 2002; Scheu et al., 2005). However, the role of the microbiota in the soil food
web is not fully understood, since many eukaryotic organisms can be considered as meta-organisms, which
carry their "own microbiome" that itself is essential for life supporting functions. From this, it is unclear if
environmental microbiomes and host specific microbiomes complement one another. So far there are no
comprehensive methods (especially not those addressing microbial functions) or standards available to



169 assess this problem. A future considering may employ the use of stable isotope labelling of select carbon

170 sources as a promising approach to follow food webs and degradation pathways (e.g. Coban et al., 2015;

171 Traugott et al., 2013).

172 Methods to assess the biodegradation of pollutants (**Function 3**), as described above, are already

173 implemented into ISO guidelines (Table 2) and are part of legal frameworks including pesticide directives

174 (EU Regulation1107/2009/EC; European Commission, 2009). A number of standard methods for the

175 determination of the potential of soils to degrade organic chemicals (**Function 3**) under both aerobic (ISO

176 14239, 2017) and anaerobic (ISO 15473, 2002) conditions are available. This emphasizes that in the past,

177 the development of standard methods was mainly driven by the need to assess the ecotoxicological effects

178 of anthropogenic activities, such as chemical contamination of soils, rather than to describe and

179 understand the natural properties and functions of soils. However, defining methods for the

180 determination of adverse effects of contaminants on soil biota was not only being done in ISO, but it was

181 also a major task of other organizations such as the Organization for Economic Co-Operation and

182 Development OECD). For example, there are OECD guidelines, tests No. 216 and 217, for testing the long-

183 term effects of single exposure chemicals on soil microbial nitrogen and carbon transformation,

184 respectively (OECD, 2000a; 2000b). As a result, it was decided early that the standardization of methods

185 for toxicity testing would not be the primary aim of the ISO sub-committee (ISO TC 190/SC 4).

186 Some of the existing standard methods that are listed in Table 2 focus on the estimation of enzyme

187 activities useful for soil quality assessment, which mainly contribute to **Function 4**. Here, the potential

188 dehydrogenase activity measurement is an indicator for general (potential) oxidoreductase activity in soil.

189 Since this measurement has been frequently used, there are large amounts of baseline data available on

190 the toxic effects of a range of pollutants in soil. Recently, additional potential enzyme activities related to

191 the C, N, P and S cycle have been used and are either standardized or are in the process for standardization.




The current evolution of molecular methods in microbial ecology has resulted in a large number
of new endpoints. It is well known that many of the new endpoints (e.g. using quantitative real-time PCR
(qPCR)) are more sensitive than classical methods that had been standardized in the past (Ribbons et al.,
2016; Schulz et al., 2016). This new metagenomics approach will be of high importance in the future, as it
allows for the implementation of information on new functional traits that can be standardized into an
analytical pipeline. For the assessment of new methods linked to **Functions 4 to 8**, qPCR from soil DNA
extracts (ISO 17601, 2016) plays a very important role in determining the abundance of single marker gene
sequences, which are indicative of specific transformation processes or soil functions. For example, the
quantification of nitrogen fixing microbes, nitrifiers and denitrifiers has been successfully implemented
using the *nifH*, *amoA* and *nirS/nirK* genes as markers, respectively (Henry et al., 2004; Hirsch et al., 2010;
Ollivier et al., 2010; Sessitsch et al., 2006). Similarly, the quantification of microorganisms involved in the
β-ketoadipate pathway has been implemented by targeting *pcaH* (El Azhari et al., 2008) and *catA* (El Azhari
et al., 2010) gene sequences. Various methods for the assessment of soil microbial **Function 4** (nutrient
cycling), **Function 5** (pest control and plant growth promotion) and **Function 7** (greenhouse gas emissions)
are proposed based on the qPCR analysis of gene sequences coding for enzymes which trigger the
respective function (e.g. Fish et al., 2013; Ribbons et al., 2016; Smith and Osborn, 2009). Additionally, it
should be noted that molecular methods based on the assessment of specific marker genes for estimating
the degradation potential in soil have already been proposed both for PAHs (e.g. Cebron et al., 2008) and
individual pesticides (e.g. Martin-Laurent et al., 2004). These could be interesting for future
standardization; however, if a method is very compound-specific and targeted, this could limit its
application range. Thus, these specific approaches will not be discussed further in this article.
Major advantages of qPCR assays to quantify gene sequence numbers, which can be used as
proxies for a given microbial process, are that they are: (i) highly standardized, sensitive, selective and
reproducible, (ii) designed for high throughput analysis, (iii) available for a wide range of targets, and (iv)



methods that are relatively cheap once the necessary analytical devices are on hand. Some training on the
method is required, however, once trained the assays are easy to perform. For example, numerous studies
have already used the microbial functional genes involved in nitrogen cycling to determine the status and
to assess induced changes in the soil microbial community (Levy-Booth et al., 2014; Nannipieri and Eldor,
2009; Wallenstein et al., 2006). Consequently, the number of functional genes that are suited for use as
specific indicators of soil function are continuing to grow in the literature as researcher gain experience in
this field and data becomes more prevalent.
Disadvantages, on the other hand, are that: (i) the quality of qPCR data depends on soil DNA
extracts (PCR inhibition), (ii) primer pairs, even degenerated ones, might not successfully amplify all
microbes of the functional group of interest, (iii) only genetic potential is resolved, and (iv) there is no
differentiation between active, dormant or dead microorganisms, when working with DNA as a template
for the qPCR reaction. The analysis of total RNA and of mRNA, which could help to overcome the latter
problem, is currently not a suitable alternative as it is highly dynamic in time and space and needs special
care to stabilize the RNA extracted from complex environmental matrices to avoid its degradation.
Another problem of DNA analysis is the biological representativeness of the results is solely based on a
relatively small amount of soil (from few hundred mg to ten g of soil) from which the DNA extracted. The
use of small soil samples (< 1 g) simplifies the sample preparation process for molecular biologists;
however, it provides a poor representation of the indigenous soil microbial community in the naturally
inhomogeneous soil. Typically, the $\alpha$-biodiversity declines with sample size while that of $\beta$-biodiversity
increases (Nicol et al., 2003; Penton et al., 2016). Lastly, it must be noted that the high repeatability and
reproducibility of molecular biology methods, including qPCR assays, depends on extraction, purification
and amplification of DNA or RNA. This is typically performed using commercial extraction kits; however,
by simply changing the commercial supplier of a kit can substantially change the results (Brooks et al.,

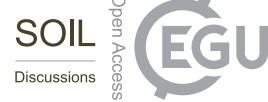

2015; Feinstein et al., 2009). This clearly challenges standardization since standard methods must not
hinge on a specific supplier.

Recently, successful examples of microbial phosphorous turnover (**Function 4)** have been

published (Bergkemper et al., 2016) where metagenomics data have been used for the construction of
primers for P mineralization, transport and uptake. As another example, the relevance of anaerobic
ammonium oxidation (anammox) for N cycling in soils has increased (Levy-Booth et al., 2014) along with
the development of analytical methods for high throughput analysis. Among the microorganisms in soil
that substantially govern pest control and plant growth promotion (**Function 5**), the most numerous
organisms are the arbuscular mycorrhizae and ectomycorrhizal fungi. These microorganisms are especially
abundant in the rhizosphere (Hartmann et al., 2009; Hayat et al., 2010; Lugtenberg and Kamilova, 2009).
Methods related to **Function 5** are listed in Table 3.

Several options exist for (additional) standardized methods to test **Function 6** (carbon cycling and

sequestration) (Table 3). For **Function 6**, there is a need to implement more fungal activity analysis as most
tests described this far only assess bacterial activities. Thus, the integration of more fungal enzyme
activities into the suite of standardized methods for soil quality assessment is essential (for example
determining the turnover of complex natural compounds such as lignin) (Baldrian, 2006). The ligninolytic
enzymes laccase and Mn-peroxidase, as well as the chitin degrading 1,4-β-*N*-acetylglucosaminidase, are
typical fungal enzymes of interest for ecosystem services (Jiang et al., 2014; Šnajdr et al., 2008). However,
since other organisms also produce these enzymes, including bacteria and plants (Bollag, 1992; de Gonzalo
et al., 2016), current methods do not specifically target fungal enzyme activities. As a result, the
implementation of molecular methods for assessing fungal communities are far less developed than those
for bacterial communities (Table 3).

The method of community level physiological profiling (CLPP) using the Biolog™ system (Biolog,

Hayward CA, USA) was first developed in the late 1980s to identify bacteria of clinical importance by



assessing the consumption of 95 different carbon sources in a microtiter plate. The technique was then
extended to identify bacterial strains from environmental mixed microbial communities samples using
select carbon sources (Garland, 1997). Currently, the technique is frequently used to assess the effects of
contaminants on soil microbial activity (Bloem and Breure, 2003; Schmitt et al., 2004). As such, the CLPP
method has become a measure of microbial functional diversity in soil (e.g. Gomez et al., 2006) and was
used to distinguish the biodiversity of soil microbial communities in monitoring programs (Rutgers et al.,
2016). Even though the method is easy to use, it does have some drawbacks (Winding and Hendriksen,
2007). The technique is based on the utilisation of select carbon sources, which when consumed result in
reduction, and thus colour change, of a tetrazolium indicator dye (Garland and Mills, 1991). This reaction
is based on the dehydrogenase enzyme activity of cultivable, fast growing, aerobic, eutrophic
microorganisms (largely bacteria). Consequently, this technique does not reflect the full spectrum of
microbial species within a mixed soil community. Additionally, due to the artificial growth conditions
required in the test, it is argued that the method does not reflect the microbial community diversity and
its function of a given soil (Glimm et al., 1997). On the other hand, however, standardized conditions allows
for direct comparisons between microbial communities in different sites, for example, independent of the
abiotic conditions, thus making CLPP a popular method for toxicology testing (Preston-Mafham et al.,

2002).

Isothermal micro-calorimetry is another technique that involves the direct measurement of

energetics in soil and provides a functional link between energy flow and the composition of belowground
microbial communities at a high taxonomic level (Herrmann et al., 2014). With this method, an integrative
determination of the metabolic activity of soil bacteria and fungi is achieved. The integrated assessment
of substances' and energy turnover has high potential to elucidate the regulation of soil ecological
functions. However, the substantial costs for the acquisition of this very specific instrumentation is
considered as a major drawback. Furthermore, the measurement requires water saturation of the soil and,



thus, the samples are modified. Since calorimetry has been rarely used and data and publications are few,
this method is considered not ready for standardization.
The methods targeting thymidine or leucine incorporation into microbial biomass can be used to
determine microbial growth rates and efficiencies (Bååth et al., 2001; Rousk, 2016). Growth rate is a
fundamental reference for numerous other microbial properties and functions. For example, it is required
to calculate microbial carbon use efficiency (CUE) as a key-parameter describing C-substrate turnover and
storage in soil (Liu et al., 2018; Spohn et al., 2016; Takriti et al., 2018). Furthermore, the method can be
used to assess the adverse effects of toxic chemicals on the microbial community (Modrzyński et al., 2016;
Rousk et al., 2009a). The drawbacks of these two methods are: (i) specific training is required, (ii)
laboratories must have a permit to manipulate radioactive isotopes, and (iii) there are higher costs for
proper handling and disposal of $^3$H-labelled radioactive material. As an alternative, the incorporation of
the stable isotope $^{18}$O from labelled water into soil microbial DNA can be used to distinguish growing and
non-growing microorganisms based on the gradient-separation of [$^{18}$O]DNA and [$^{16}$O]DNA (Schwartz,
2007). The $^{18}$O stable isotope method has been improved by sequencing a marker gene from fractions
retrieved from ultracentrifugation to produce taxon density curves; thus enabling researchers to estimate
the percent isotope composition of each microbial taxon's genome (Schwartz et al., 2016). This method
continues to be advanced and, although not used often, could have a high potential for future
standardization.
There are simplistic methods available to determine organic matter decomposition which are
indicative of C cycling (**Function 6**). The tests listed in Table 3 are based on measuring the weight loss of
introduced organic materials of different complexity in soil over time. The tests are relatively easy to
perform and inexpensive, however, degradation activity is not exclusive to microorganisms but can also
include invertebrates. The OECD litter bag test (OECD, 2006) for site specific assessment of organic matter
decomposition uses wheat straw as the substrate and provides clear evidence of cellulose degradation. In





general, the litter bag tests provides evidence for the degradation of naturally occurring plant material in
soil. Results do, however, depend on the mesh size of the litter bags (increasing exclusion of soil animals
with decreasing mesh size). On the other hand, plant material or litter is hard to standardise with the
results largely depending on the composition of the plant material. As such, artificial cellulose has been
successfully used for a laboratory procedure to assess organic matter decomposition (Kvas et al., 2017).
Another alternative to the litter bag test is the use of tea bags (Keuskamp et al., 2013). Tea bags can be
purchased to contain a consistent quality of material, and so this method is preferred by citizen science
(e.g. farmers to assess the soil quality of their land). In order to better distinguish the degrading abilities
of different soil microbiomes, the test could be modified to use different types of tea that contain
recalcitrant material to a different extent. Another test for future method development is the Bait Lamina
test (ISO 18311, 2016) used to assess the degradation of organic matter in field soil by grazing
invertebrates (Jänsch et al., 2013; Kvas et al., 2017). It is a simple test that can easily be adapted for use
under controlled laboratory conditions (Jänsch et al., 2017).

Methods for the determination and assessment of greenhouse gas emissions from soil (**Function**

**7**) have already been standardized or are well advanced in the standardization process (Table 2). They are
mostly focused on measuring concentrations of greenhouse gases, like $CO_2$, $CH_4$ and $N_2O$, as well as their
fluxes as endpoints. In addition, molecular biology methods that estimate the relative abundance of
functional microbial guilds or taxa gives new insight into the ecology of microorganisms involved in the
formation of greenhouse gases. For example, the qPCR measurement of key $N_2O$ functional genes has
allowed researchers to link $N_2O$ reduction capacity to reduced greenhouse gas emissions in soil amended
with organic matter (Xu et al., 2018). Additionally, the quantification of functional gene sequences related
to methane generation and methane oxidation, respectively, yields detailed insights into the functional
potential of climate change-affected permafrost soils (Yergeau et al., 2010).



For **Function 8** (soil structure affecting soil water, gas balance and filtration function), there is clear
evidence that microbial activity and biomolecules substantially contribute to the formation and stability
of micro-aggregates, and thus to the structure, pore system and pre-consolidation stress of soils (Six et al.,
2004). While existing parameters, such as enzyme activities, are not clearly indicative in this regard (Beck
and Beck, 2000), glomalin can be considered as a biochemical marker of soil aggregation. This glycoprotein
is produced by microorganisms, especially arbuscular mycorrhiza fungi, and significantly increases
aggregate formation and stability (Rillig, 2004; Rillig and Mummey, 2006). The existing protocols for
extraction (chemical extraction combined with autoclaving) and determination of glomalin, either by using
the Bradford protein assay, enzyme-linked immunosorbent assay (ELISA), or LC-MS method (Bolliger et al.,
2008; Janos et al., 2008), open the possibility for its standardization in the near future. It should be noted,
however, that a well-equipped and experienced laboratory is required to perform this method.

**4 Transforming standardized methods into indicators of soil quality**
As recently underlined by the European Food Safety Agency (EFSA) in a scientific opinion 'addressing the
state of the science on risk assessment of plant protection products for in-soil organisms', there is an
urgent need to modernize pesticide risk assessment by implementing specific protection goals for in-soil
organisms which are key drivers of a wide range of functions supporting ecosystem services (Ockleford et
al., 2017). There currently exists a multitude of methods that can potentially be used for this task. Here,
we have identified in the body of this paper a number of methods that are presumably suitable for further
evaluation and standardization with regard to their scientific value and practical applicability. These
prospective standardized methods will not only be useful to identify adverse effects on the soil
microbiome, but also to conduct comparable studies in laboratories all over the world to define normal
operating ranges of microbial activity in soil and respective quality indices and threshold values.



It is clear that all parameters taken together reflect the potential of a microbial community to
perform a certain function and not solely a specific (actual) activity. This is important to understand to
interpret the values of a given endpoint in relation to both energy fluxes and compound transformation
rates, which can largely depend on intrinsic properties such as pedoclimatic conditions, and nutrient
availability as well as extrinsic properties such as anthropogenic effects, and soil cultivation measures. To
make use of these methods as indicators for soil quality, there are several requirements that need to be
included. These involves the assessment of the normal operating range of soil that include natural and
dynamic fluctuations of a given endpoint. The methods need to be implemented into a framework, which
takes into account site-specific conditions including soil type, pedoclimate and land-use. Additionally,
there is a requirement for the assessment of resistance and resilience of a given microbial endpoint to see
how much it is affected by a soil disturbance and whether or not it can recover (e.g. return to its original
state) after the disturbance has disappeared. Also, the use of a test battery to measure a range of
interconnected endpoints is recommended (Ockleford et al., 2017) to integrate different biological and
other parameters (e.g. soil pH, organic carbon content) into multiparametric indices (Bastida et al., 2008;
Kvas et al., 2017). Finally, to fully understand soil microbial functioning, a task was envisioned to
investigate the linkage between the genetic functional potential and the available resources, termed the
soil metaphenome (Jansson and Hofmockel, 2018). This will require even further integration and
assessment of multiple parameters and test methods. Reaching that goal will surely promote soil
ecological research but, at the current stage, may clearly go beyond the applied aim of standardization to
release easy-to-use targeted methods.
The critical evaluation of existing and non-standardized methods is required to further select and
standardize new methods to assess soil quality. For methods linked to the molecular analysis of soil
microbiomes; there is a need to ensure that worldwide activities are synchronized to propose important
standards that are well accepted by the scientific community. For example, recently, the Earth Microbiome





Project (www.earthmicrobiome.org) has proposed primer pairs to barcode soil bacteria in a standardized
manner. Furthermore, new bioinformatic pipelines have been developed that are being used more and
more as standard procedures. Finally, to improve the reproducibility of data it has been agreed that a
complex mixture of microorganisms (MOC) must be implemented as a control in every experiment. The
exact composition of the MOC is still under discussion, however, it is clear that if further developments of
microbial bar coding and/or metagenomics methods are to be implemented into ISO guidelines, an MOC
is required.

ISO standardization committees are open circles and the presented selection and valuation of

methods may not be complete. Environmental scientists are solicited to propose new work items enlarging
the current catalogue of biological methods for future standardization. Accordingly, this opinion paper
aims at initiating a broader discussion intended to improve the measurement of microbial functions for
soil quality assessment.


**Competing interests.** The authors declare that they have no conflict of interest.

**Acknowledgements.** The compilation of methods was developed during the meetings of the ISO TC 190/SC
4 working group 4 "Soil biological methods". We thank our colleagues for all the fruitful discussions. We
gratefully thank Lily Pereg for reviewing a first draft of this manuscript and we will keep her in our
remembrance. Biochem-Env is a service of the investment d'Avenir infrastructure AnaEE-France, overseen
by the French National Research Agency (ANR) (ANR-11-INBS-0001).

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





**Table 1.** List of criteria for the selection of indicators for microbial functional indicators, based on Faber et
al. (2013) and Pulleman et al. (2012), with slight modifications by the authors.

| | Criteria | Measured by | Low Score | High Score |
|---|---|---|---|---|
| **a)** | Practicability | Lab equipment | Very few labs have the equipment needed | All labs would be able to carry out the work |
| | | Skills | Specialist skills are needed | General skills would suffice |
| **b)** | Cost efficiency | Capital start-up | More than €100 000 | Less than €2000 |
| | | Cost per sample | More than €100 | Less than €2 |
| | | Labour needed in the lab | High labour demand | Low labour demand |
| | | Labour needed in the field | High labour demand | Low labour demand |
| **c)** | Policy relevance | Focus on ecosystem processes and services | Weak links with existing or planned legislation | Strong links with existing or planned legislation |
| **d)** | Sensitivity | Effect of soil properties | No response or idiosyncratic response | The indicator responds characteristically to change |
| | | Effect of land use | No response or idiosyncratic response | The indicator responds characteristically to change |
| | | Effect of disturbance | No response or idiosyncratic response | The indicator responds characteristically to change |
| **e)** | Selectivity | | Endpoint affected by numerous variables | Endpoint only affected by parameter under investigation |
| **f)** | Reproducibi-lity | | Low or largely varying reproducibility among replicates | Highly reproducible |
| **g)** | Use as an indicator | Status quo | Not in use already | In use already |
| **h)** | Handling and availability of organisms[1] | | Rare and/or difficult to obtain Difficult to keep Largely varying quality/fitness Seasonal availability | Easy to obtain Easy to keep Easy to provide with constant quality/fitness Year-round availability |
| **i)** | Fit for use as an indicator | Significance / explanatory power | Weak relationship to ecological function | Strong relationship to ecological function |
| | | Standardized | Methods are not ready for general use or standardization (i.e. low experience, no SOPs [2]) | Methods are already in general use, preferably as standard (e.g. OECD) |
| | | Spatio-temporally relevant | Spatio-temporally only relevant for a small plot at one point in time | Representative for more than one site and/or more than one point in time |
| | | Understandable | Difficult to explain in a policy situation | Easily understood in a policy situation |



| | | | | |
|---|---|---|---|---|
| **j)** | Experience | Literature data | Low amount of information on performance and outcome, e.g. <10 publications | High amount of information on the performance and outcome, e.g. >10 publications, existing ring test(s) |
| **k)** | Data evaluation | Database | No or hardly any existing data available or not freely available | Freely available and sound database for data evaluation |

[1] Only relevant for faunal species. Does not apply to soil microorganisms that are tested with their natural
abundance in mixed communities.
[2] Standard operating procedures





**Table 2.** Methods already validated and published as ISO standards for determining potential microbial
biomass and activities for soil quality.

| **Microbial biomass and respiration (some relations to Functions 1 and 6)** | |
|---|---|
| ISO 14240-1 | Determination of soil microbial biomass – Part 1: Substrate induced respiration method |
| ISO 12240-2 | Determination of soil microbial biomass – Part 2: Fumigation – extraction method |
| ISO 16072 | Laboratory method for determination of microbial soil respiration |
| ISO 17155 | Determination of the activity of the soil microflora using respiration curves |
| ISO 11063 | Direct soil DNA extraction |
| ISO 17601 | Quantification of the abundance of microbial groups in soil DNA extract |
| ISO/TS 29843-1 | Method by phospholipid fatty acid analysis (PLFA) and phospholipid ether lipids (PLEL) analysis ( |
| ISO/TS 29843-2: | Method by phospholipid fatty acid analysis (PLFA) using the simple PLFA extraction method |
| **(Potential) microbial enzymatic activities: C, N and P turnover (Functions 4 and 6)** | |
| ISO/TS 22939 [1] | Measurement of enzyme activity patterns in soil samples using fluorogenic substrates in micro-well plates |
| ISO/DIS 20130 EN[2] | Measurement of enzyme activity patterns in soil samples using colorimetric substrates in micro-well plates |
| ISO/TS 23753-1 | Determination of dehydrogenase activity in soils — Part 1: Method using triphenyltetrazolium chloride (TTC) |
| ISO/TS 23753-2 | Determination of dehydrogenase activity in soils — Part 2: Method using iodotetrazolium chloride (INT) |
| ISO 14238 | Biological methods – Determination of nitrogen mineralization and nitrification in soils and the influence of chemicals on these processes |
| ISO 15685 | Determination of potential nitrification and inhibition of nitrification — Rapid test by ammonium oxidation |
| **Potential microbial activities: biodegradation of pollutants (Function 3)** | |
| ISO 11266 | Guidance on laboratory testing for biodegradation of organic chemicals in soil under aerobic conditions |
| ISO 14239 | Laboratory incubation systems for measuring the mineralization of organic chemicals in soil under aerobic conditions |
| ISO 15473 | Guidance on laboratory testing for biodegradation of organic chemicals in soil under anaerobic conditions |
| **Potential microbial activities: turnover greenhouse gases (Function 7)** | |
| ISO/DIS 20951 | Guidance on methods for measuring greenhouse gases ($CO_2$, $N_2O$, $CH_4$) and ammonia ($NH_3$) fluxes between soils and the atmosphere |
| ISO/TS 20131-1 | Easy laboratory assessments of soil denitrification, a process source of $N_2O$ emissions -- Part 1: Soil denitrifying enzymes activities |
| ISO/TS 20131-2 | Easy laboratory assessments of soil denitrification, a process source of $N_2O$ emissions -- Part 2: Assessment of the capacity of soils to reduce $N_2O$ |
| **Potential microbial activities: organic matter decomposition (Function 6)** | |
| ISO/CD 23265 [3] | Test for measuring organic matter decomposition in contaminated soil |

[1] Measured enzyme activities: Arylsulfatase E.C. 3.1.6.1; α-glucosidase E.C. 3.2.1.20; β-glucosidase E.C.
3.2.1.21; β-xylosidase E.C. 3.2.1.37; cellobiosidase E.C. 3.2.1.91; N-acetylglucosaminidase E.C. 3.2.1.52;



841 phosphodiesterase E.C. 3.1.4.1; phosphomonoesterase E.C. 3.1.3.2; leucine-aminopeptidase E.C.

842 3.4.11.1; alanine-aminopeptidase E.C. 3.4.11.12.

843 [2] Measured enzyme activities: Arylamidase E.C. 3.4.11.2; arylsulfatase E.C. 3.1.6.1; $\alpha$-glucosidase E.C.

844 3.2.1.20; $\beta$-glucosidase E.C. 3.2.1.21; $\beta$-galactosidase E.C. 3.2.1.22; N-acetylglucosaminidase E.C.

845 3.2.1.52; phosphatase E.C. 3.1.4.1; acid phosphatase E.C. 3.1.4.1; alkaline phosphatase E.C. 3.1.4.1;

846 urease E.C. 3.5.1.5.

847 [3] Degradation of cellulose under laboratory conditions.





**Table 3.** Potential new methods for the ISO standardization process and assessment according to the "logical sieve" selection criteria (described in Table 1).

| Method | Source | Function addressed | Assessment [1] | | | | | | | | | | |
|---|---|---|---|---|---|---|---|---|---|---|---|---|---|
| | | | a | b | c | d | e | f | g | h | i | j | k |
| **Function 4. Nutrient cycling (N and P)** | | | | | | | | | | | | | |
| Functional genes assessed by real time qPCR | | | | | | | | | | | | | |
| Ammonium monoxygenase gene (*amoA*) | Levy-Booth et al., 2014 | quantify the abundance of nitrifying microbes | 1-2 | 3 | 5 | 5 | 5 | 5 | 5 | na[2] | 4 | 4 | 3 |
| Ammonium monoxygenase gene (*amoB*) | Norton et al., 2002 | quantify the abundance of nitrifying microbes | 1-2 | 3 | 5 | 5 | 5 | 5 | 5 | na | 4 | 4 | 3 |
| Nitrogenase gene (*nifH*) | Gaby and Buckley, 2012 | quantify the abundance of N fixing microbes | 1-2 | 3 | 5 | 5 | 5 | 5 | 5 | na | 4 | 4 | 3 |
| Various genes driving P turnover | Bergkemper et al, 2016 | quantify the abundance of microbes driving P transformation | 1-2 | 3 | 5 | 5 | 5 | 5 | 5 | na | 4 | 4 | 3 |
| **Function 5. Pest control and plant growth promotion** | | | | | | | | | | | | | |
| Specific mtDNA sequences assessed by real time qPCR | Voříšková et al., 2017 | quantify the abundance of arbuscular mycorrhiza | 1-2 | 3 | 5 | 5 | 5 | 5 | 5 | na | 4 | 3 | 2 |
| Specific ITS sequences assessed by real time qPCR | Sakakibara et al., 2002 | quantify the abundance of ectomycorrhizal fungi | 1-2 | 3 | 5 | 5 | 5 | 5 | 5 | na | 4 | 3 | 2 |
| Specific ITS sequences assessed by real time qPCR | Savazzini et al., 2008 | quantify the abundance of biocontrol active Trichoderma fungi | 1-2 | 3 | 5 | 5 | 5 | 5 | 5 | na | 4 | 3 | 2 |
| **Function 6. Carbon cycling and sequestration** | | | | | | | | | | | | | |
| Enzyme activity of fungi | Eichlerová et al., 2012 | determine activity of laccases | 4 | 4 | 5 | 5 | 3 | 5 | 5 | na | 4 | 5 | 4 |
| | Bach et al., 2013 | determine activity of phenoloxidases | 4 | 4 | 5 | 5 | 3 | 5 | 5 | na | 4 | 5 | 4 |
| Community level physiological profiling (CLPP, "Biolog") | Garland and Mills, 1991 | determine degradation of a set of carbon sources | 3 | 4 | 3 | 1 | 1 | 5 | 3 | na | 1 | 5 | 3 |
| Microcalorimetry | Prado and Airoldi, 2001; 2003 | quantify microbial energy turnover | 1 | 2 | 1 | 3 | 3 | 5 | 1 | na | 3 | 3 | 1 |
| [³H]-leucine or [³H]-thymidine incorporation | Bååth, 1998; Bååth et al., 2001; Rousk et al., 2009b | quantify microbial growth rate and efficiency | 1 | 2 | 5 | 5 | 4 | 5 | 5 | na | 4 | 4 | 2 |
| [¹⁸O] incorporation into DNA from labelled water | Schwartz, 2007; Schwartz et al., 2016 | quantify microbial growth rate and efficiency | 2 | 3 | 5 | 5 | 4 | 5 | 5 | na | 3 | 2 | 2 |
| Organic matter decomposition | OECD, 2006; Knacker et al., 2003 | assess organic matter degradation and therefore C cycling | 5 | 5 | 4 | 4 | 5 | 4 | 5 | na | 5 | 5 | 5 |
| Litter bag technique | Bockhorst and Wardle, 2013 | assess the degradation of plant litter material | | | | | | | | | | | |
| Tea bag technique | Keuskamp et al., 2013 | assess the degradation of tea leaves | 5 | 5 | 4 | 4 | 5 | 4 | 5 | na | 5 | 5 | 5 |





**Table 3.** Continued

| Method | Source | Function addressed | Assessment [1] | | | | | | | | | | | |
|---|---|---|---|---|---|---|---|---|---|---|---|---|---|---|
| Funct. genes within C cycle assessed by real time qPCR | El Azhari et al., 2008 | quantify the abundance of microbes able to degrade protocatechuate (*pcaH*) a key intermediary metabolite of the β-ketoadipate pathway | 1-2 | 3 | 5 | 5 | 5 | 5 | 5 | na | 4 | 4 | 3 | |
| | El Azhari et al., 2010 | quantify the abundance of microbes able to degrade catechol (*cat A*) a key intermediary metabolite of the β-ketoadipate pathway | | | | | | | | | | | | |
| **Function 7. Greenhouse gas emissions** | | | | | | | | | | | | | | |
| Methyl coenzyme M reductase (*mcrA*) assessed by real time qPCR | Steinberg and Regan, 2009 | quantify the abundance of methane producing microbes | 1-2 | 3 | 5 | 5 | 5 | 5 | 5 | na | 4 | 4 | 3 | |
| N$_2$O reductase gene (*nosZ*) assessed by real time qPCR | Jung et al., 2013 | quantify the abundance of N$_2$0 reducing microbes | 1-2 | 3 | 5 | 5 | 5 | 5 | 5 | na | 4 | 4 | 3 | |
| Methane reductase gene *pmoA*) | Kolb et al., 2003 | quantify the abundance of methane reducing microbes | 1-2 | 3 | 5 | 5 | 5 | 5 | 5 | na | 4 | 4 | 3 | |
| Nitric oxide reductase gene *cnorA*) assessed by real time qPCR | Dandie et al., 2007 | quantify the abundance of methane reducing microbes | 1-2 | 3 | 5 | 5 | 5 | 5 | 5 | na | 4 | 4 | 3 | |
| **Function 8. Soil structure affecting soil water, gas balance and filtration capacity** | | | | | | | | | | | | | | |
| Determination of glomalin | Bolliger et al., 2008; Janos et al., 2008; Wright et al., 1998 | determine the content of glomalin in soil as a proxy of soil aggregation | 3 | 3 | 4 | 4 | 4 | 5 | 3 | na | 4 | 4 | 2 | |

[1] Overall scoring in case of several measures for one criterion. Fulfilment of criterion described by
numbering (colour code): 1 (red) very low; 2 (orange) low; 3 (yellow) medium; 4 (light green) good; 5
(dark green) very good.
[2] na = not applicable