# Peer review of "Identification of new microbial functional standards for soil quality assessment"

_SOIL, 2019_

## Referee Comment (RC1) · Robert Griffith (Referee) · 5 Aug 2019

This paper discusses new methodologies and opportunities offered by molecular methodologies to provide microbiological indicators for assessing soil quality. In essence, it reports on key issues raised during recent discussions within the International Organization for Standardization (ISO), and identifies the need to focus on soil functions of relevance to ecosystem services as recognised for example in the Millennium Ecosystem Assessment. A key focus of the paper is in highlighting and scoring the potential of qPCR approaches for quantifying functional gene abundances of relevance to providing simple metrics relevant for quantification of biogeochemical fluxes (which are difficult to quantify directly).

The paper is generally well written, interesting, and delivers on synthesising the current broad status with respect to these issues, and additionally proposes some potentially new indicator approaches which could be implemented. As such, I feel it makes a useful contribution. The paper could be improved by offering more critical analyses of the approaches; as well as authoritatively defining the new science needed to facilitate the implementation of more robust soil microbiological indicators.

Three areas which could be elaborated on further I feel are highlighted below. Perhaps fully covering them in detail extends beyond the remit of this manuscript, so I leave it to the editor to decide whether they should be expanded upon in the article (alternatively I guess these publically viewable comments may constitute a contribution to the "discussion" format of the journal. . ..).

1. Indicator targets within the global soil geographic context. The paper briefly mentions this on line 365 ("methods need to be implemented into a framework, which takes into account site-specific conditions"), but offers no specific ways forward for this critical issue. Are elevated abundances of a functional indicator always "desirable", and how might indicator target values, and indeed the indicators themselves differ for different soil systems? I'm not sure if we even have a good soil classification system or framework that allows us to set regionalised targets for the simple variable of soil carbon, and I sense this is what causes pushback on soil targets from industry and policymakers. Given this, could proposing even more microbiological variables be deemed somewhat premature?

2. Relatedly, what is the evidence that gene abundance relates to functions of relevance to ecosystem services? It is often stated that you cannot infer anything about processes from gene abundance alone, but I feel there is little literature actually specifically addressing this with robust contrasts within an ES indicator context. For example comparative data for ammonia oxidation gene abundances does actually appear to relate to nitrification rates in certain studies, so do we need a critical meta-analyses of this now for a variety of indicators? Again, relating to the point above, do we always

want high nitrification, high litter decomposition, high enzyme activity etc in all soil systems; and is there any evidence that molecular detection of elevated pathogens reliably informs on plant health...Essentially what do these measures really tell us about desirable ES outcomes, and if there is little information available, then what can be done to progress?

3. Standardisation: essential for policy, but bad for science? Given the paper's policy focus, it appears to heavily endorse standardisation. However molecular ecology is a rapidly growing field, and technologies change (eg sequencing platforms) which causes issues with implementing standardised protocols. Scientific developments must be free to progress in order to develop the deep and often complex understanding of processes required to implement meaningful process indicators. It would be useful to highlight this potentially conflicting issue...

---

## Referee Comment (RC2) · Emilia Hannula (Referee) · 16 Sep 2019

The topic of relating currently available measurement techniques to the soil functions is a very timely issue. Using logical sieve approach, this article investigates the suitability of commonly used methods to evaluate the soil functions and ecosystem services, one at the time. The paper is very clearly written and presents solid, interesting research. It could benefit from section on future directions and more updated list of currently available (molecular) techniques. Will these be better or do we already have a golden standard that will tell us all we want to know?

It is clear that the work has started already in 2013 and in the past six years huge developments have been achieved in the toolbox available to measure soil functions and

especially diversity. Authors mention the 'Earth microbiome project' and standardization of primer sets and pipelines to study bacterial diversity as an emerging technique. However, in reality, this method is the most used method to study fungi and bacteria in soils and is considered fairly standard as all labs use the same regions (V4 and ITS2) and often same primers. There is a recent article on the methodological comparison on bacteria (Ramirez et al. 2018, Detecting macroecological patterns in bacterial communities across independent studies of global soils, Nature microbiology). There is much discussion on using 'mock' (not MOC)-communities to standardize the methods and most labs use these already.

Furthermore, the field is moving towards true (shotgun) metagenomics sequencing which yields data on all soil organisms and their functions. This approach is emerging but will definitely be worth discussing in this context. It will have no bias of PCR but the quantity of soil used and DNA extraction efficiency related issues will remain. In short, the 'newer' methods should be discussed. For the methods used to study diversity and future avenues in that field, following paper can be cited (Geisen et al. 2019, A methodological framework to embrace soil biodiversity, SBB).

In the abstract, it is mentioned that especially fungal functions are difficult to measure. The article presents a suite of measures traditionally used and recommends to use enzyme production to measure fungal functions. For bacteria, qPCR based methods are recommended. This discrepancy in recommendations should be discussed. Why is it not feasible in all case to look at process rates (i.e. decomposition), but details on the amount of enzymes and/or amount of organisms performing the task are needed? In which scenarios and which scales which measurement is needed? Considering this would make the article stronger and bring more to the field. In the evaluation of function 6 (carbon cycling) it is simply stated that because these enzymes are often produced by other organisms, molecular methods are less developed. This is partly true but can be related also that the enzyme measurements are pretty good and give the actual rate of enzyme measured. Furthermore, there are existing primer sets for quite some

of the CAZys (for example: Edwards et al. 2011: Simulated Atmospheric N Deposition Alters Fungal Community Composition and Suppresses Ligninolytic Gene Expression in a Northern Hardwood Forest, PlosOne Gorfer et al. 2011: Community profiling and gene expression of fungal assimilatory nitrate reductases in agricultural soil, ISMEJ Chen et al. 2013: Comparative analysis of basidiomycetous laccase genes in forest soils reveals differences at the cDNA and DNA levels. Plant and Soil Hannula & van Veen (2016) Primer Sets Developed for Functional Genes Reveal Shifts in Functionality of Fungal Community in Soils. Frontiers in Microbiology These measurements from DNA have problems as different fungi have different copy numbers/ types of genes and not everything that is in the DNA is expressed. Indeed, work is needed to get these methods ISO certified but more discussion on the future directions would be welcome.

---

## Author Comment (AC1) · 29 Nov 2019

As the corresponding author and on behalf of all coauthors of the manuscript soil-2019-42 "Identification of new microbial functional standards for soil quality assessment" I herewith resubmit a revised version of the manuscript. First of all we want to express our appreciation of the very constructive and sound suggestions and criticism of both reviewers. Following these advice, we significantly revised the manuscript and added text passages. Additionally we did some further polishing in style and language. Please find our response to the reviewers' suggestions in the following text (marked as text in italics). Reviewer #1 Robert Griffith This paper discusses new methodologies and opportunities offered by molecular methodolo-gies to provide microbiological indicators for assessing soil quality. In essence, it reports on key issues raised during recent discussions within the International Organization for Stand-ardization (ISO), and identifies the need to focus on soil functions of relevance to ecosystem services as recognised for example in the Millennium Ecosystem Assessment. A key focus of the paper is in highlighting and scoring the potential of qPCR approaches for quantifying functional gene abundances of relevance to providing simple metrics relevant for quantifica-tion of biogeochemical fluxes (which are difficult to quantify directly).

The paper is generally well written, interesting, and delivers on synthesising the current broad status with respect to these issues, and additionally proposes some potentially new indicator approaches which could be implemented. As such, I feel it makes a useful contribu-tion. The paper could be improved by offering more critical analyses of the approaches; as well as authoritatively defining the new science needed to facilitate the implementation of more robust soil microbiological indicators.

We especially added further aspects and discussions on soil microbial methods, which also relate to the points raised by reviewer # 2, E. Hannula.

Three areas which could be elaborated on further I feel are highlighted below. Perhaps fully covering them in detail extends beyond the remit of this manuscript, so I leave it to the editor to decide whether they should be expanded upon in the article (alter-natively I guess these publically viewable comments may constitute a contribution to the "discussion" format of the journal. . ..). 1. Indicator targets within the global soil geographic context. The paper briefly mentions this on line 365 ("methods need to be implemented into a framework, which takes into account site-specific conditions"), but offers no specific ways forward for this critical issue. Are elevat-ed abundances of a functional indicator always "desirable", and how might indicator target values, and in-deed the indicators themselves differ for different soil systems? I'm not sure if we even have a good soil classification system or framework that allows us to set regional-ised targets for the simple variable of soil carbon, and I sense this is what causes pushback on soil targets from industry and policymakers. Given this, could proposing even more micro-biological variables be deemed somewhat premature?

This is a very good point and emphasizes the need to gather a database of soil mi-crobial pa-rameters in worldwide soils (with links to soil chemical, physical and pedo-climatic data) in or-der to get some idea about 'normal' value ranges. We fully agree that 'higher' values do not necessarily mean 'better' and such typical value ranges are needed to identify unusual aberra-tions. However, this fundamental discussion would lead a bit beyond the scope of this paper, which is focused on the methods that are re-quired to receive the results, independent from how we store and assess these results. Consequently and pointing into the direction raised by R. Griffiths we added on lines 397-399: "Undoubtedly, this requires further joint efforts in order to generate compre-hensive databases from which normal operating ranges of values for a given proxy can be read. Such a task calls for standardized methods to obtain comparable data".

2. Relatedly, what is the evidence that gene abundance relates to functions of rele-vance to ecosystem services? It is often stated that you cannot infer anything about processes from gene abundance alone, but I feel there is little literature actually specif-ically addressing this with robust contrasts within an ES indicator context. For example comparative data for am-monia oxidation gene abundances does actually appear to relate to nitrification rates in cer-tain studies, so do we need a critical meta-analyses of this now for a variety of indicators? Again, relating to the point above, do we always want high nitrification, high litter decomposi-tion, high enzyme activity etc in all soil systems; and is there any evidence that molecular detection of elevated pathogens re-liably informs on plant health...Essentially what do these measures really tell us about desirable ES outcomes, and if there is little information availa-ble, then what can be done to progress?

We share the view of the reviewer that we are only at the beginning to understand the meaning of microbial activity and functional gene abundance in terms of ecosystem services and func-tioning of soils. The following phrases of the added text mostly apply to this comment. 250-256 "Also, evidence is increasing that functional gene abun-dance and community struc-ture are closely linked to related microbial activities and

their increase or decrease, e.g. through agricultural fertilizer regime or soil contamina-tion (Levy-Booth et al., 2014; Ouyang et al., 2018; Xue et al., 2018). However, also contrasting findings have been reported, pointing to the fact that functional gene abun-dance and diversity is less affected by short-term changes, e.g. due to soil moisture changes (Zhang et al., 2019). A critical meta-analysis of existing data and reports, respectively, would be timely to better identify and generalize the linkage of func-tional gene abundance and ecosystem services." 402-408: "Here the use of DNA based methods, which provide a measure of a microbial com-munity's potential to perform a given process, might be more useful than using RNA. The RNA rather indicates actual activities, which may highly fluctuate in time and space, and thus are of less signifi-cance as an indicator. However, free DNA released from dead microbes is often highly resistant in soil, which might result in an over estimation of a potential function. This needs to be taken into account when interpreting the data. Recently, methods that extract DNA only from living cells have been described, but their use has not been yet introduced into re-cent standardization activities."

3. Standardisation: essential for policy, but bad for science? Given the paper's pol-icy focus, it appears to heavily endorse standardisation. However molecular ecology is a rapidly growing field, and technologies change (eg sequencing platforms) which causes issues with imple-menting standardised protocols. Scientific developments must be free to progress in order to develop the deep and often complex understanding of processes required to implement meaningful process indicators. It would be useful to highlight this potentially conflicting is-sue.

We agree that standardization is a balancing act. We aim to receive largely comparable data (asking for defined methods) but want to use the latest methods for that purpose (asking for highest flexibility). So we added on lines 429-436: "Lastly, it must be noted that standardization of methods is inevitably a balancing act. On one hand, standard-ization provides defined meth-ods that are essential to obtain comparable data, e.g. for integration in large, joint databases. On the other hand, it requires setting a specific

method for several years. Consequently, scien-tific progress cannot be easily adopted, or at least with a delay, considering that standards are revised every five years, which may be a barrier to the introduction of new approaches result-ing from technological evolution, especially in the fast developing field of molecular biology methods. Hence, it is also the aim of this paper to have an open discussion to identify the best suitable methods with an assumed longer period of validity."
* * *

---

## Author Response (AR1)

Universität Trier  D-54286 Trier

Mrs. Natascha Töpfer
Copernicus Publications
Editorial Support
on behalf of the SOIL Editorial Board

Prof. Dr. Sören Thiele-Bruhn
Abteilung Bodenkunde, FB VI

☎ +49 (0)651/201-2241
Fax        201-3809
Sekr.      201-2242
E-Mail thiele@uni-trier.de

November 2012

**MS No.: soil-2019-42; MS Type: Forum article**

Dear Natascha Töpfer, dear editorial board members,

As the corresponding author and on behalf of all coauthors of the manuscript soil-2019-42 "*Identification of new microbial functional standards for soil quality assessment*" I herewith re-submit a revised version of the manuscript. First of all we want to express our appreciation of the very constructive and sound suggestions and criticism of both reviewers. Following these advice, we significantly revised the manuscript and added text passages. Additionally we did some further polishing in style and language.

Please find our response to the reviewers' suggestions in the following text (marked as text in italics).

I very much hope that the revised version, which we think was substantially improved, finds your and the reviewers approval.

With kind regards,

S. Thiele-Bruhn

Reviewer #1 Robert Griffith

This paper discusses new methodologies and opportunities offered by molecular methodologies to provide microbiological indicators for assessing soil quality. In essence, it reports on key issues raised during recent discussions within the International Organization for Standardization (ISO), and identifies the need to focus on soil functions of relevance to ecosystem services as recognised for example in the Millennium Ecosystem Assessment. A key focus of the paper is in highlighting and scoring the potential of qPCR approaches for quantifying functional gene abundances of relevance to providing simple metrics relevant for quantification of biogeochemical fluxes (which are difficult to quantify directly).

The paper is generally well written, interesting, and delivers on synthesising the current broad status with respect to these issues, and additionally proposes some potentially new indicator approaches which could be implemented. As such, I feel it makes a useful contribution. The paper could be improved by offering more critical analyses of the approaches; as well as authoritatively defining the new science needed to facilitate the implementation of more robust soil microbiological indicators.

*We especially added further aspects and discussions on soil microbial methods, which also relate to the points raised by reviewer #2, E. Hannula.*

Three areas which could be elaborated on further I feel are highlighted below. Perhaps fully covering them in detail extends beyond the remit of this manuscript, so I leave it to the editor to decide whether they should be expanded upon in the article (alternatively I guess these publically viewable comments may constitute a contribution to the "discussion" format of the journal. . ..).

1. Indicator targets within the global soil geographic context. The paper briefly mentions this on line 365 ("methods need to be implemented into a framework, which takes into account site-specific conditions"), but offers no specific ways forward for this critical issue. Are elevated abundances of a functional indicator always "desirable", and how might indicator target values, and indeed the indicators themselves differ for different soil systems? I'm not sure if we even have a good soil classification system or framework that allows us to set regionalised targets for the simple variable of soil carbon, and I sense this is what causes pushback on soil targets from industry and policymakers. Given this, could proposing even more microbiological variables be deemed somewhat premature?

*This is a very good point and emphasizes the need to gather a database of soil microbial parameters in worldwide soils (with links to soil chemical, physical and pedoclimatic data) in order to get some idea about 'normal' value ranges. We fully agree that 'higher' values do not necessarily mean 'better' and such typical value ranges are needed to identify unusual aberrations. However, this fundamental discussion would lead a bit beyond the scope of this paper, which is focused on the methods that are required to receive the results, independent from how we store and assess these results. Consequently and pointing into the direction raised by R. Griffiths we added on lines 397-399: "Undoubtedly, this requires further joint efforts in order to generate comprehensive databases from which normal operating ranges of values for a given proxy can be read. Such a task calls for standardized methods to obtain comparable data".*

2. Relatedly, what is the evidence that gene abundance relates to functions of relevance to ecosystem services? It is often stated that you cannot infer anything about processes from gene abundance alone, but I feel there is little literature actually specifically addressing this with robust contrasts within an ES indicator context. For example comparative data for ammonia oxidation gene abundances does actually appear to relate to nitrification rates in certain studies, so do we need a critical meta-analyses of this now for a variety of indicators? Again, relating to the point above, do we always want high nitrification, high litter decomposition, high enzyme activity etc in all soil systems; and is there any evidence that molecular detection of elevated pathogens reliably informs on plant health...Essentially what do these measures really tell us about desirable ES outcomes, and if there is little information available, then what can be done to progress?

*We share the view of the reviewer that we are only at the beginning to understand the meaning of microbial activity and functional gene abundance in terms of ecosystem services and functioning of soils. The following phrases of the added text mostly apply to this comment.*
*250-256 "Also, evidence is increasing that functional gene abundance and community structure are closely linked to related microbial activities and their increase or decrease, e.g. through agricultural fertilizer regime or soil contamination (Levy-Booth et al., 2014; Ouyang et al., 2018; Xue et al., 2018). However, also contrasting findings have been reported, pointing to the fact that functional gene abundance and diversity is less affected by short-term changes, e.g. due to soil moisture changes (Zhang et al., 2019). A critical meta-analysis of existing data and reports, respectively, would be timely to better identify and generalize the linkage of functional gene abundance and ecosystem services."*
*402-408: "Here the use of DNA based methods, which provide a measure of a microbial community's potential to perform a given process, might be more useful than using RNA. The RNA rather indicates actual activities, which may highly fluctuate in time and space, and thus are of less significance as an indicator. However, free DNA released from dead microbes is often highly resistant in soil, which might result in an over estimation of a potential function. This needs to be taken into account when interpreting the data. Recently, methods that extract DNA only from living cells have been described, but their use has not been yet introduced into recent standardization activities."*

3. Standardisation: essential for policy, but bad for science? Given the paper's policy focus, it appears to heavily endorse standardisation. However molecular ecology is a rapidly growing field, and technologies change (eg sequencing platforms) which causes issues with implementing standardised protocols. Scientific developments must be free to progress in order to develop the deep and often complex understanding of processes required to implement meaningful process indicators. It would be useful to highlight this potentially conflicting issue.

*We agree that standardization is a balancing act. We aim to receive largely comparable data (asking for defined methods) but want to use the latest methods for that purpose (asking for highest flexibility). So we added on lines 429-436: "Lastly, it must be noted that standardization of methods is inevitably a balancing act. On one hand, standardization provides defined methods that are essential to obtain comparable data, e.g. for integration in large, joint databases. On the other hand, it requires setting a specific method for several years. Consequently, scientific progress cannot be easily adopted, or at least with a delay, considering that standards are revised every five years, which may be a barrier to the introduction of new approaches*

*resulting from technological evolution, especially in the fast developing field of molecular biology methods. Hence, it is also the aim of this paper to have an open discussion to identify the best suitable methods with an assumed longer period of validity."*

Reviewer #2 Emilia Hannula
The topic of relating currently available measurement techniques to the soil functions is a very timely issue. Using logical sieve approach, this article investigates the suitability of commonly used methods to evaluate the soil functions and ecosystem services, one at the time. The paper is very clearly written and presents solid, interesting research. It could benefit from section on future directions and more updated list of currently available (molecular) techniques. Will these be better or do we already have a golden standard that will tell us all we want to know?

*For sure we don't have the golden standard (otherwise new method development would be obsolete). We added further text and discussion especially on soil molecular biology methods and parameters.*
*200-210: metagenomics for degrading activities.*
*218-226: Problems with data handling and bioinformatics, respectively.*
*250-256: Discussion on linkage between functional gene abundance and function.*

It is clear that the work has started already in 2013 and in the past six years huge developments have been achieved in the toolbox available to measure soil functions and especially diversity. Authors mention the 'Earth microbiome project' and standardization of primer sets and pipelines to study bacterial diversity as an emerging technique. However, in reality, this method is the most used method to study fungi and bacteria in soils and is considered fairly standard as all labs use the same regions (V4 and ITS2) and often same primers. There is a recent article on the methodological comparison on bacteria (Ramirez et al. 2018, Detecting macroecological patterns in bacterial communities across independent studies of global soils, Nature microbiology). There is much discussion on using 'mock' (not MOC)-communities to standardize the methods and most labs use these already.

*Discussion on barcoding using high throughput sequencing and in special the use of primer pairs targeting the V4 region of the 16S rRNA gene and ITS2 region for bacterial and fungal barcoding was added to the text. Also we evaluate the suitability of bioinformatics pipelines for cross comparison. (Lines 157-167).*

Furthermore, the field is moving towards true (shotgun) metagenomics sequencing which yields data on all soil organisms and their functions. This approach is emerging but will definitely be worth discussing in this context. It will have no bias of PCR but the quantity of soil used and DNA extraction efficiency related issues will remain. In short, the 'newer' methods should be discussed. For the methods used to study diversity and future avenues in that field, following paper can be cited (Geisen et al. 2019, A methodological framework to embrace soil biodiversity, SBB).

*Metagenomics are further discussed now on lines 218-226. We thank for the helpful literature reference that was included in the new added text.*

In the abstract, it is mentioned that especially fungal functions are difficult to measure. The article presents a suite of measures traditionally used and recommends to use enzyme production to measure fungal functions. For bacteria, qPCR based methods are recommended. This discrepancy in recommendations should be discussed. Why is it not feasible in all case to look at process rates (i.e. decomposition), but details on the amount of enzymes and/or amount of organisms performing the task are needed? In which scenarios and which scales which measurement is needed? Considering this would make the article stronger and bring more to the field. In the evaluation of function 6 (carbon cycling) it is simply stated that because these enzymes are often produced by other organisms, molecular methods are less developed. This is partly true but can be related also that the enzyme measurements are pretty good and give the actual rate of enzyme measured. Furthermore, there are existing primer sets for quite some of the CAZys (for example: Edwards et al. 2011: Simulated Atmospheric N Deposition Alters Fungal Community Composition and Suppresses Ligninolytic Gene Expression in a Northern Hardwood Forest, PlosOne Gorfer et al. 2011: Community profiling and gene expression of fungal assimilatory nitrate reductases in agricultural soil, ISMEJ Chen et al. 2013: Comparative analysis of basidiomycetous laccase genes in forest soils reveals differences at the cDNA and DNA levels. Plant and Soil Hannula & van Veen (2016) Primer Sets Developed for Functional Genes Reveal Shifts in Functionality of Fungal Community in Soils. Frontiers in Microbiology These measurements from DNA have problems as different fungi have different copy numbers/ types of genes and not everything that is in the DNA is expressed. Indeed, work is needed to get these methods ISO certified but more discussion on the future directions would be welcome.

*Fungi: Information on methods targeted towards fungi was added and the recommended literature included. Fct. 6. The specific problem of linking copy numbers of fungal genes to the size of a functional population was emphasized (lines 291-297).*

*Decomposition rates: We present and propose molecular methods (being an indirect measure of activities, rather looking on the organism side) but also on activities such as the litter bag and the tea bag method, thus targeting much more the effect point of view (see Table 3). We also added new text on the discussion in how far functional gene abundance can serve as a measure of true soil microbial functioning (lines 250-256).*

*Scenarios and scales: We agree that a bundle of methods is proposed with differences regarding application range and purpose (despite the function represented). To discuss this, would lead much deeper into the criteria (Table 1) d) sensitivity (with effect of soil properties, of land use and of disturbance on the test result), e) selectivity, and g) use as an indicator. We feel that this topic is surely relevant but would require too much detail information and discussion, making the manuscript too voluminous. We decided to leave the manuscript more concise and hope to find the reviewers' consent for that.*

[revised manuscript text omitted]